# Impaired FADD/BID signaling mediates cross-resistance to immunotherapy in Multiple Myeloma

Umair Munawar[1], Xiang Zhou[1], Sabrina Prommersberger[1], Silvia Nerreter[1], Cornelia Vogt[1], Maximilian J. Steinhardt [1], Marietta Truger[2], Julia Mersi[1], Eva Teufel[1], Seungbin Han[1], Larissa Haertle [1,3], Nicole Banholzer[4], Patrick Eiring[4], Sophia Danhof [1], Miguel Angel Navarro-Aguadero[3], Adrian Fernandez-Martin[3], Alejandra Ortiz-Ruiz [3], Santiago Barrio[3,5], Miguel Gallardo [3], Antonio Valeri[3], Eva Castellano[3], Peter Raab[6], Maximilian Rudert[6], Claudia Haferlach[2], Markus Sauer [3], Michael Hudecek [1], J. Martinez-Lopez[4,5], Johannes Waldschmidt[1], Hermann Einsele [1], Leo Rasche [1] & K. Martin Kortüm [1✉]

The treatment landscape in multiple myeloma (MM) is shifting from genotoxic drugs to immunotherapies. Monoclonal antibodies, immunoconjugates, T-cell engaging antibodies and CART cells have been incorporated into routine treatment algorithms, resulting in improved response rates. Nevertheless, patients continue to relapse and the underlying mechanisms of resistance remain poorly understood. While Impaired death receptor signaling has been reported to mediate resistance to CART in acute lymphoblastic leukemia, this mechanism yet remains to be elucidated in context of novel immunotherapies for MM. Here, we describe impaired death receptor signaling as a novel mechanism of resistance to T-cell mediated immunotherapies in MM. This resistance seems exclusive to novel immunotherapies while sensitivity to conventional anti-tumor therapies being preserved in vitro. As a proof of concept, we present a confirmatory clinical case indicating that the *FADD/BID* axis is required for meaningful responses to novel immunotherapies thus we report impaired death receptor signaling as a novel resistance mechanism to T-cell mediated immunotherapy in MM.

[1] Department of Internal Medicine II, University Hospital of Würzburg, Würzburg, Germany. [2] MLL Munich Leukemia Laboratory, Munich, Germany. [3] Department of Hematology, Hospital Universitario 12 de Octubre, CNIO, Complutense University Madrid, Madrid, Spain. [4] Department of Biotechnology and Biophysics, University of Würzburg, Würzburg, Germany. [5] Altum Sequencing Co., Madrid, Spain. [6] Department of Orthopaedic Surgery, König Ludwig Haus, University of Würzburg, Würzburg, Germany. ✉email: Kortuem_m@ukw.de

Multiple myeloma (MM) is the second most common hematologic malignancy and is characterized by monoclonal plasma cell expansion in the bone marrow (BM). The disease is fatal for most patients and cure is only achieved in a minor subset. The treatment for multiple myeloma is changing dramatically from conventional therapies, such as chemotherapy and radiation[1] towards novel immunotherapies including monoclonal antibodies, antibody-drug conjugates (ADC), T-cell engaging antibodies (TCE) and CART cells (CARTs) giving substantial hope to considerably improve the prognosis in MM[2]. However, not all patients benefit equally from such innovative therapies, and prognosis remains poor for patients at relapse or even primary resistance.

Various resistance mechanisms have been described to immunotherapy in MM which include sub-clonal diversity[3], altered tumor microenvironment[4], loss of target antigen[5], point mutations within the target antigen[6], T-cell exhaustion[7] or activation of immune checkpoint pathways including the PD-L1/PD-1 axis[8]. Still, in most cases, the reason for suboptimal treatment response or relapse remains unexplained and further mechanisms need to be explored.

One of the hallmarks of drug resistance in cancer cells is the evasion from apoptosis induction[9]. This has been demonstrated for classical chemotherapies i.e., paclitaxel, doxorubicin, and cisplatin[10]. Interestingly, dysfunction of apoptotic pathway signaling has recently been linked to anti-CD19 CART cell resistance in acute lymphoblastic leukemia (ALL)[11]. One of the key players of extrinsic apoptotic pathway is the Fas-associated protein with death domain (FADD). FADD plays a pivotal role in apoptosis by activating (pro-) caspases leading to BID cleavage[12] resulting in apoptosis. Dysregulation of FADD has been reported to be involved in cancer progression[13].

To study the role of impaired apoptosis pathway regulation for immunotherapy resistance in MM, we generated MM cell models with impaired death receptor signaling. Failure of T-cell mediated cytotoxicity was observed in our models whereas treatment response against conventional cytotoxic therapy was preserved. Our in vitro findings were corroborated by clinical data proposing that the FADD/BID axis is indispensable for meaningful responses to novel immunotherapies in vivo.

## Results

### FADD and BID knock-out impairs the apoptotic machinery in MM cells.
To investigate the role of death receptor signaling in MM resistance to immunotherapy, we performed CRISPR-Cas9 based knockout of two candidate genes (FADD and BID) in two independent MM cell lines (AMO1, L363). KO clones were screened for gene expression with Taqman Real-Time PCR assays (Invitrogen) and positive KOs were confirmed with Sanger sequencing. We confirmed the functional impact of the KO genes on apoptotic activity using an apoptosis-inducing, CD95-activating antibody (anti-Fas Ab). Caspase-3/7 activity was increased in wild type (WT) cells to $1.5 \times 10^6$ RFU compared to $0.3 \times 10^6$ RFU in FADD[KO] cells compared with relevant control cells after treatment with anti-Fas Ab for 24 h ($p < 0.0001$). Similarly, Caspase-8 activity was much higher in WT cells $1.0 \times 10^5$ RFU compared to FADD[KO] clones with $0.2 \times 10^4$ RFU ($p = 0.0004$).

In BID[KO] cells, only a minor increase in levels of caspases ($7.6 \times 10^5$ RFU and $6.6 \times 10^5$ RFU) was observed compared to WT cells ($1.5 \times 10^6$ RFU) ($p < 0.0004$, $p = 0.0075$ Fig. 1a, b). Furthermore, no induction of apoptosis was detected in FADD[KO] cells after treatment with anti-Fas Ab compared to WT cells where around 57% apoptotic cells were seen via Annexin-V PI staining (Fig. 1c, d). Western blot analysis detected the cleaved

PARP in WT cells and its absence in AMO1[KO] clones (Fig. 1e, Supplementary Fig. S4), confirming that our MM[KO] models had an impaired apoptotic pathway machinery.

### FADD / BID axis regulates resistance to T-cell mediated immunotherapy and is dispensable for conventional anti-MM therapies.
To validate the significance of FADD/BID gene expression in T-cell based immunotherapies, we treated our cells with the CD38-directed monoclonal antibody daratumumab, which is commonly used in first-line treatment of MM. We observed significantly reduced killing activity of daratumumab in our apoptosis-impaired MM models compared to WT. In the AMO1 cell line 38.3% killing efficiency in WT was reduced to 21.2% in the case of FADD[KO] and 23.2% in BID[KO] clones ($p < 0.05$, Fig. 2a) Likewise, in L363 cells, we observed higher mean killing efficacy in WT cells (66.5%) which was reduced to 41.9% in FADD[KO] cells ($p = 0.0014$, Fig. 2b).

To further substantiate on this finding, we next investigated the efficacy of BCMA-CART killing of MM[KO] cells. BCMA-CARTs mediated a striking killing efficiency of around 92.1% of WT AMO1 cells which was significantly reduced to around 43.6% in the case of FADD[KO] clones ($p < 0.0001$) and to 43.3% in BID[KO] cells ($P = 0.015$, Fig. 2c). Similar impact was observed in L363 cell line where killing efficacy in WT cells was reduced from 64.1% in WT cells to 27.0% in FADD[KO] clones and 78.3% in WT cells to 56.4% in BID[KO] clones ($p < 0.05$, Fig. 2d).

Teclistamab, a FDA-approved anti-CD3 x anti-BCMA T-cell engaging antibody, was tested in our knock-out model as an additional T-cell mediated immunotherapy. We found a significant reduction in the activity of teclistamab in FADD[KO] cells compared to respective WT clones as percentage of surviving cells was increased from 32.4% in WT to 74.3% in KO clones ($p = 0.04$, Fig. 2e).

We further tested other approved anti-MM therapeutics such as genotoxic drugs (doxorubicin, melphalan), a proteasome inhibitor (PI, bortezomib) and an immunomodulatory drug (IMiD, lenalidomide) and found no differences in their IC50 amongst WT and various KO models (Supplementary Fig. S1). Notably, the treatment efficacy with the MM approved ADC belantamab mafodotin was not altered.

In summary, these findings confirm that defects in extrinsic apoptotic circuitry conferred by FADD and BID knock-out results in reduced activity of T-cell mediated immunotherapies in MM, whereas conventional anti-MM agents remain equally effective.

### Loss of surface antigen or (CAR-)T cell dysfunction is not responsible for resistance.
To exclude loss of target antigen as the underlying cause of resistance, we used super-resolution direct Stochastic Optical Reconstruction Microscopy (dSTORM) to quantify receptor density on the tumor cell surface at the single cell level[14]. CD38 and BCMA loss of antigen were excluded as the mode of resistance, and we found no significant changes in their expression levels among WT and KO models. Furthermore, no changes in receptor density or antigen surface distribution were seen between WT and KO cells in AMO1 (Fig. 3a) or L363 cells (Supplementary Fig. S2).

The functional profile of BCMA-CARTs was determined in response to antigen stimulation with WT and KO cells. No significant differences were found in IL-2 and IFN-γ (Fig. 3b) after antigen-specific stimulation for 20 h. Furthermore, using the same method no difference in the activity of T-cells was observed after incubation with target cells and Teclistamab (Fig. 3c). These data confirm that loss of antigen or dysfunction of (CAR-)T cells is not responsible for the resistance observed.

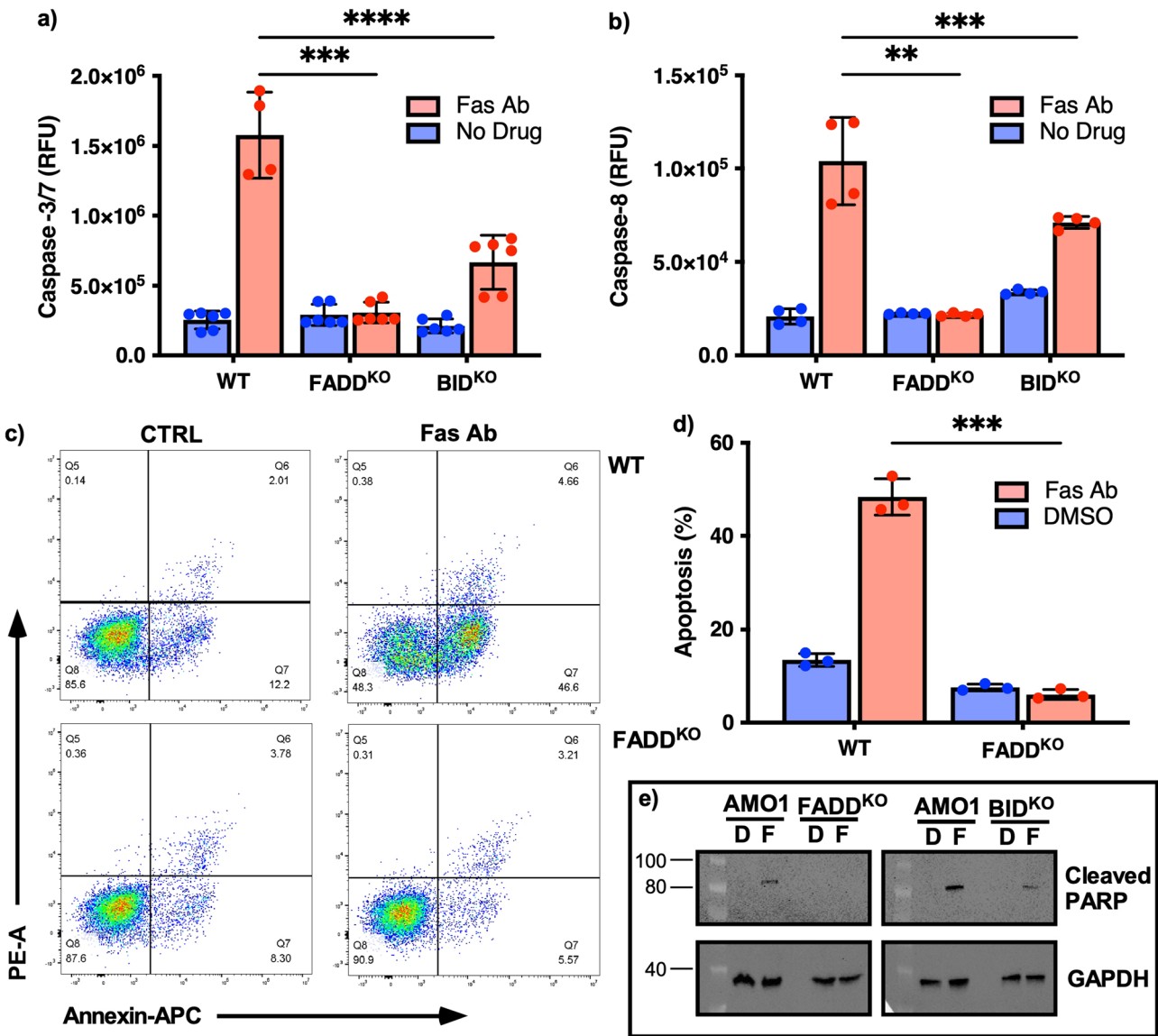

**Fig. 1 FADD and BID are two key players of the apoptotic machinery in MM cells.** Changes in levels of Caspase-3/7 (**a**) and Caspase-8 (**b**) when WT AMO1 cell line and FADD^KO and BID^KO cells were treated with 15 µg/ml anti-Fas Ab for 24 h. Annexin-V PI staining of cells after treatment with DMSO (CTRL) Fas Ab (**c**) and quantification of apoptosis (**d**). Western blot analysis of cleaved PARP upon activation of apoptotic pathway in WT and KO cells after exposition to DMSO (D) or Fas Ab (F) (**e**). *=$p < 0.05$; **=$p < 0.005$; ***=$p < 0.0005$, ****=$p < 0.0001$.

**Low expression of FADD and BID is a potential mechanism of resistance to immunotherapy in patients.** To determine the clinical relevance of our findings, we screened for expression levels of key players of apoptotic machinery in a pilot cohort of $n = 24$ newly diagnosed MM (NDMM) and $n = 53$ relapsed refractory MM (RRMM) patients without prior exposure to TCE and/or CART cell therapy. Expression levels were normalized to $n = 26$ CD138$^+$ bone marrow plasma cell samples from healthy donors. We did not find significant differences in expression levels for Caspase 3,7,8,9 and c-FLIP across cohorts in NDMM vs. RRMM patients (Supplementary Fig. S3). However, when we ranked the patients according to their gene expression of FADD and BID, the ten patients with the lowest values were predominantly daratumumab-resistant (7/10 and 9/10, respectively), whereas in the patients with highest FADD and/or BID expression daratumumab resistance was less pronounced (3/10 and 4/10, respectively, $p = 0.057$, Fig. 4a, b). Patient characteristics and clinical courses of daratumumab-resistant patients are described in Supplementary Table 1. Notably, a 60-year-old male patient

(MM1), with IgA kappa MM and high-risk features (amp1q21 and t(14;20)) progressed after not more than two cycles of Daratumumab-Pomalidomide-Dexamethasone with low expression levels of FADD and BID and was bridged with KTD-PACE to subsequent BCMA-targeted CART cell therapy, to which he was responding with a complete remission (CR, Fig. 4c). At relapse six months later, the patient could be rescued by GPRC5D-directed TCE and achieved a very good partial remission (VGPR). FADD and BID expression declined (Fig. 4d, e) under continuous therapy and the patient progressed with extra medullary disease (EMD) (Fig. 4c). He was then unresponsive to BCMA-targeted TCE, but responded to belantamab mafodotin, ruling out BCMA target loss and leaving room for speculation whether impaired death receptor signaling was a potential resistance mechanism in this patient.

## Discussion

FADD plays an important role in tumorigenesis. Its overexpression has been associated with metastasis in breast tumors

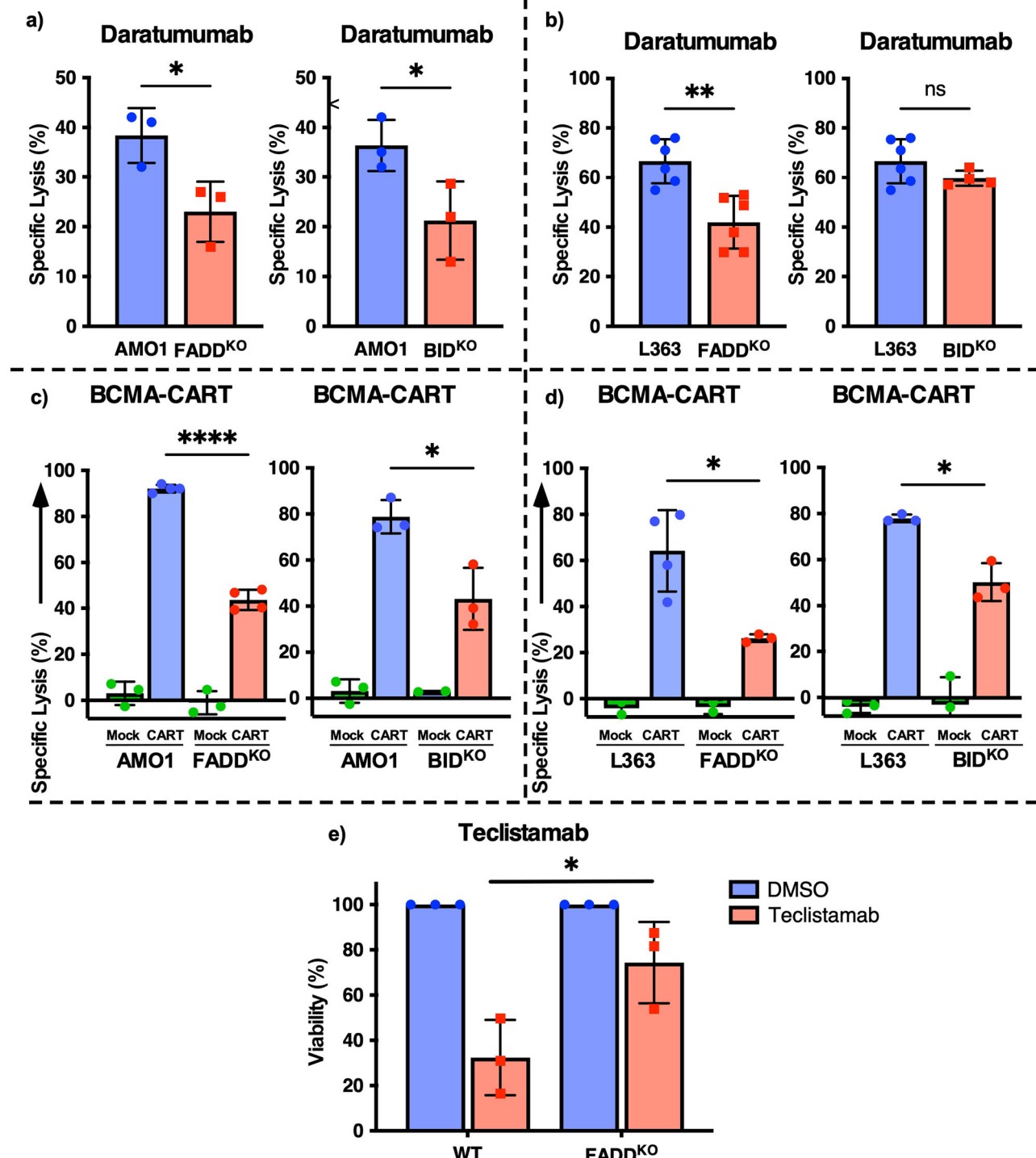

**Fig. 2 FADD/BID axis regulates T-cell mediated immunotherapies.** Luminescence-based cell survival assay for WT, FADD[KO] and BID[KO] AMO1 (**a**) and L363 (**b**) cells lines treated with daratumumab and co-cultured with PBMCs from healthy donors. Target cells (AMO1 (**c**)) (L363 (**d**)) were mixed with BCMA-CARTs or untransduced T cells (Mock) in 10:1 E:T ratio for FADD[KO] and 5:1 E:T ratio for BID[KO] cells and were subjected to the luminescence-based cell survival assay. L363 WT and FADD[KO] cells were co-cultured with Pan-T cells isolated from healthy donors in the presence of teclistamab and viability was determined by luminescence-based cell survival assay (**e**). *=$p < 0.05$; **=$p < 0.005$; ***=$p < 0.0005$.

and hence inferior clinical outcome[15] and reduced expression has been linked with development of oral squamous cell carcinoma and myelodysplastic syndrome[16,17]. *BID*, a member of the pro-apoptotic BCL2 family, contributes to chemosensitivity[18], with BID-null mouse embryonic fibroblast cells exhibiting resistance towards DNA damaging agents. Singh et al. were first to demonstrate the correlation between decreased death receptor

expression and primary resistance to CD19 CARTs in ALL[11]. Our data suggest that the dysfunction of apoptosis is also related to the failure of T-cell mediated immunotherapy in multiple myeloma. In our knock-out models activity of BCMA-CARTs, the mono-clonal antibody daratumumab and the bispecific antibody teclistamab was significantly reduced, whereas conventional chemotherapy such as melphalan and doxorubicin did not show

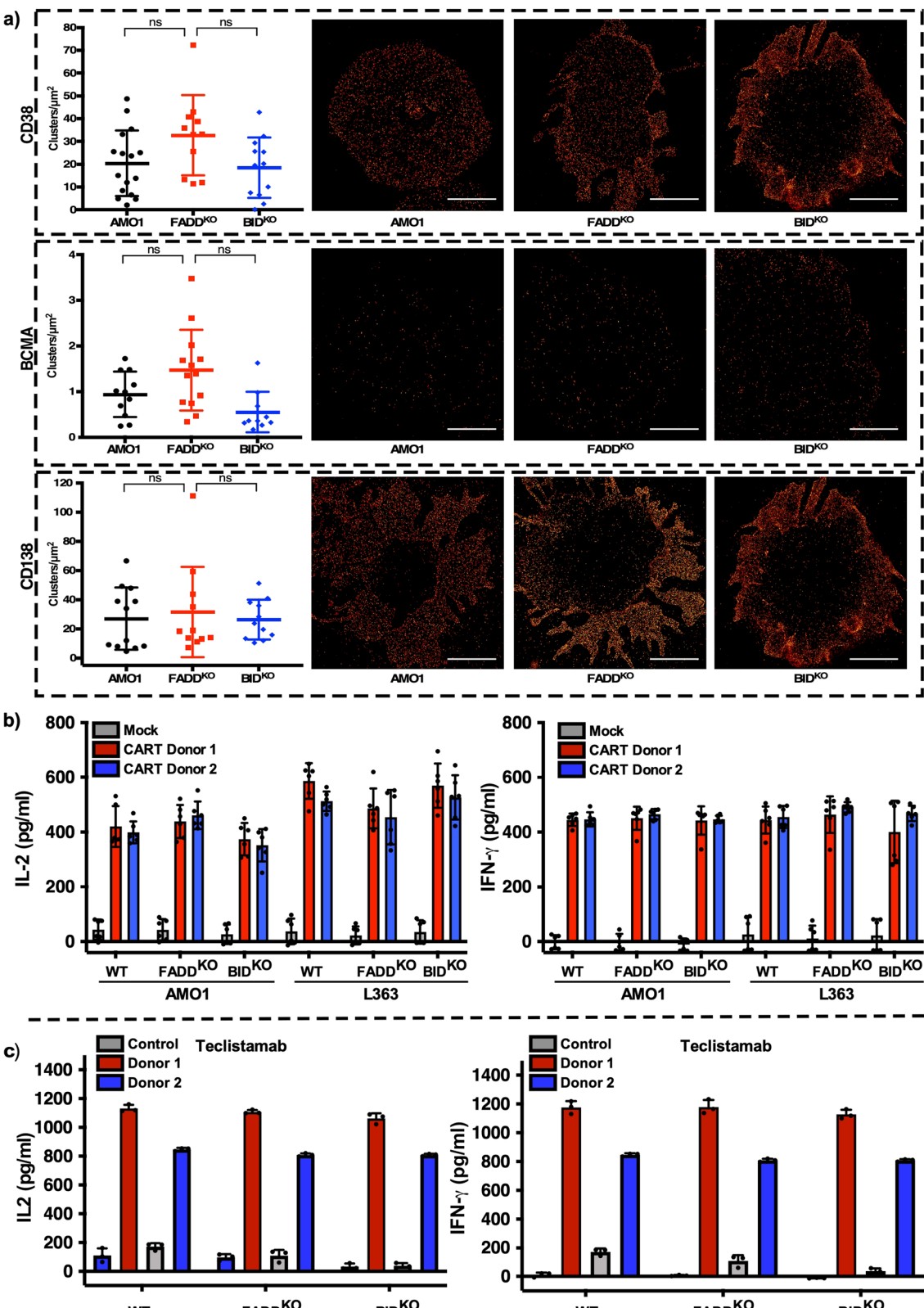

**Fig. 3 Loss of target antigen or dysfunction of (CAR-)T cells are not responsible for therapeutic resistance. a** Cells were stained for CD38 (Top panel) BCMA (middle panel) and CD138 (bottom panel) and imaged by $d$STORM (Scale bar = 5 μm). No differences in clusters/μm² for mentioned antigens was observed. BCMA-CARTs or (**b**) and Pan-T cells (**c**) from two different donors were stimulated with WT (AMO1-L363) or respective KO cells for 20 h and levels or IL-2 and IFN-γ were measured. Mock represents the untransduced T cells (**b**) and Control represents the base level of IL-2 and IFN-γ in absence of Teclistamab (**c**).

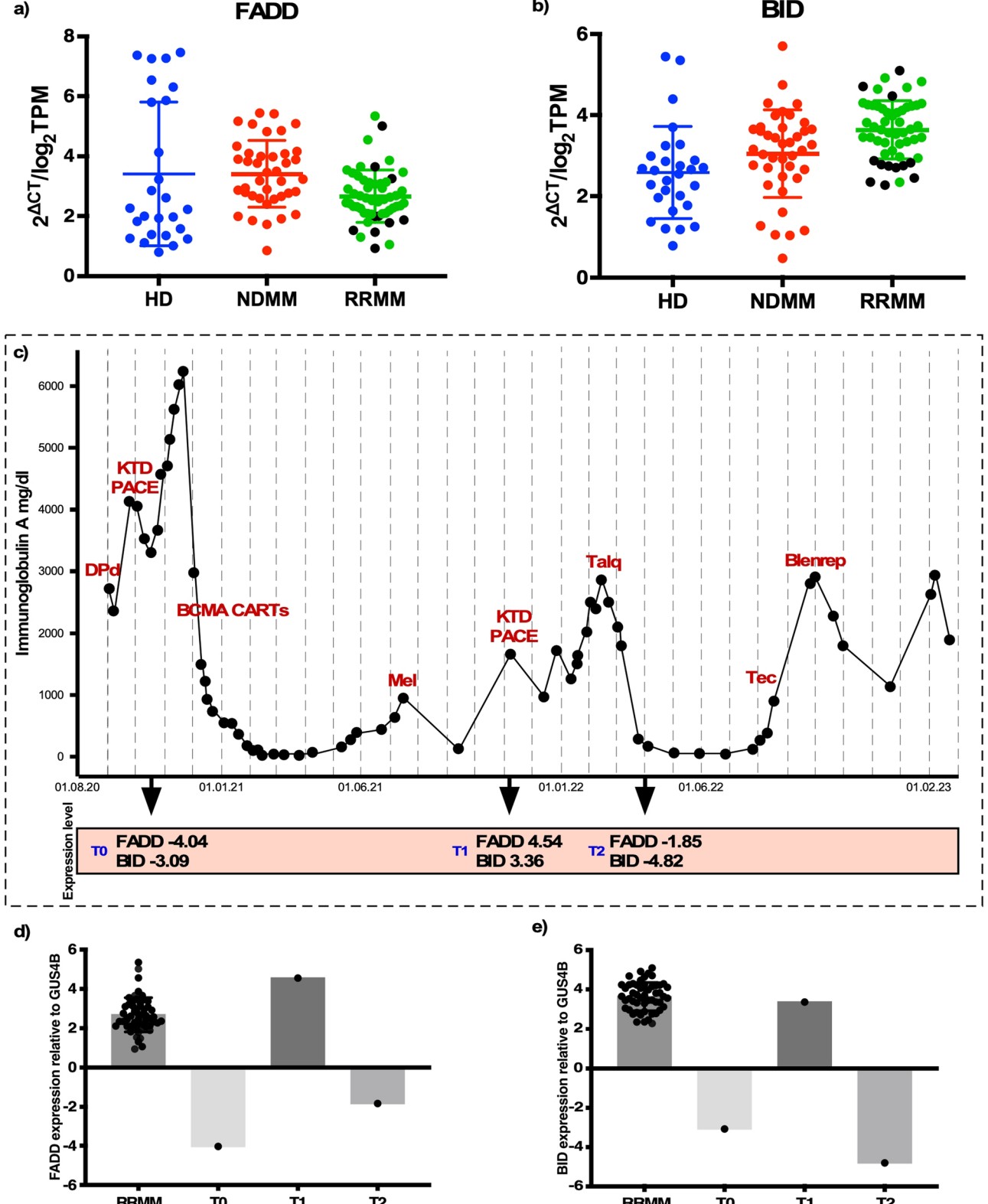

**Fig. 4 Low expression of *FADD/BID* axis is associated with therapy resistance.** Gene expression data of *FADD* (**a**) and *BID* (**b**) in healthy donors (HD), NDMM and RRMM patients. Gene expression was determined by bulk RNA seq and normalized as $\log_2$TPM and quantified by qPCR and normalized as $2^{\Delta CT}$. Daratumumab-resistant patients are marked as black dots. **c** IgA protein levels to monitor disease burden in a patient (MM1) along with treatments given overtime, response, and expression levels of *FADD* and *BID* genes. Expression level of *FADD* (**d**) and *BID* gene (**e**) relative to housekeeping gene *GUS4B* at three different time points (T0–T2) compared to mean expression level in RRMM patient cohort.

any impairment in efficacy. These drugs are known to have genotoxic effects in target cells[10] implying that the effect is caused by other alternative pathways (intrinsic) than the extrinsic apoptotic pathway.

We correlated this finding with data from a cohort of $n = 78$ MM patients. As hypothesized low expression of the apoptotic pathway genes *FADD* and *BID* was associated with inferior response to daratumumab therapy. However, in our retrospective real world analysis we did not observe differing mean levels of key apoptotic genes between NDMM vs. RRMM, most likely due to the variety of different combination therapies that the patients received. Nevertheless, in our Daratumumab resistant RRMM patients a clear trend towards low expression of FADD/BID was visible.

Apoptosis is a key process in maintaining tissue homeostasis by eliminating unwanted or damaged cells within the body. Here, we provide first evidence that apoptosis is also crucial for the mode of action of T-cell mediated immunotherapy in MM. Our study indicates that the ability of MM cells to undergo apoptosis in response to treatment is an indispensable factor for the treatment-success of T-cell mediated immunotherapy (Fig. 5).

## Methods

**Cell culture**. AMO1 and L363 were purchased from the German Collection of Microorganisms and Cell Cultures (DSMZ). Cells were cultured in standard culture medium RPMI1640 supplemented with 10% FBS, 1 mM Sodium Pyruvate, 2 mM Glutamate, 100U/ml Penicillin and 100 μg/ml Streptomycin at 37 °C in 5% ambient $CO_2$.

**CRISPR-Cas guide RNA design, generation and screening of KO clones**. Geneart CRISPR nuclease vector with OFP Reporter Kit (ThermoFisher Scientific, A21174) was used to generate plasmid for disruption of *FADD* and *BID* genes. CRISPR gRNAs

were designed using the Broad Institute GPP web portal. Oligonucleotides (Supplementary Table 2) were annealed as dsDNA fragments and cloned into CRISPR vector to create a functional gRNA expression plasmid. Neon electroporation system (ThermoFisher Scientific) was used for delivery of plasmids into the cells. Cells were sorted 48 h post-electroporation using OFP as a sorting marker via FACS Aria III (Becton-Dickinson). Individual single-cell clones were allowed to grow and screened for expression of respective genes using specific TaqMan predesigned probes (Hs00609632_m1, Hs00538709_m1) (ThermoFisher Scientific).

**Bioluminescence-based cell survival assays**. Cells were engineered for stable expression of firefly Luciferase. WT cell lines and respective KO clones were electroporated with 20 μg/ml sleeping beauty expression vector expressing Luciferase cDNA and GFP and pTX100 transposase expression plasmid. Transfected cells were subjected to puromycin selection (1 μg/ml 10–14 days) to obtain cells stably transfected with luciferase. Cell survival assays were performed by adding D-luciferin (ThermoFisher Scientific) to the cell culture and incubating at 37 °C for 5 min. Specific lysis was calculated using control of untreated cells without effectors/drugs. Cells were co-cultured with various effector:target ratios of BCMA-CARTs or control T-cells for 48 h and bioluminescence was measured with Tecan Infinite 200 pro plate reader (Tecan).

**Flow cytometry**. Cells were washed with PBS and resuspended in Annexin-V binding buffer (Immunostep), followed by incubation with 2 μl Annexin DY 634 at RT for 15 min. After washing once with PBS, cells were resuspended in 200 μl of buffer containing 2 μl PI. Apoptotic cells were determined by using cytoflex (Beckman Coulter). Data was analyzed using Flowjo 9.0. (Flowjo,

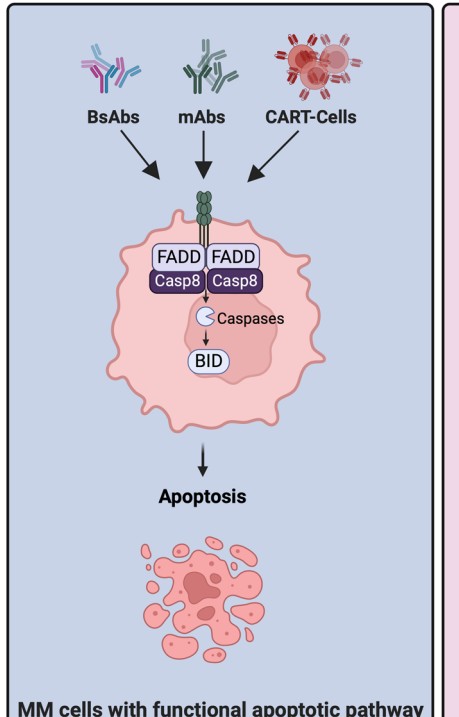
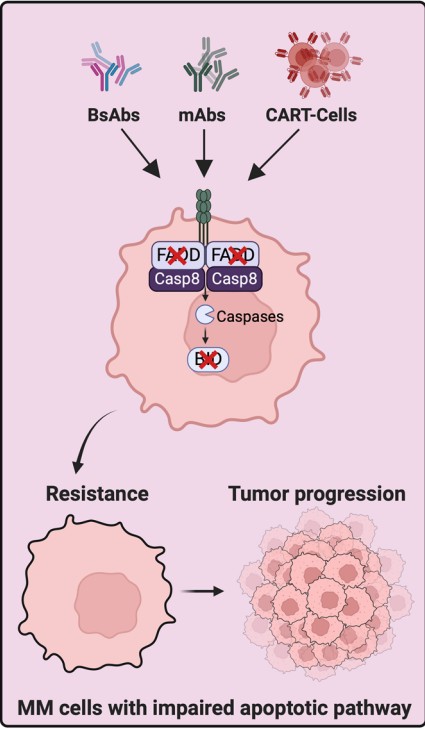

**Fig. 5 Graphical summary of the study.** MM cells with intact apoptotic machinery are sensitive to T-cell mediated immunotherapy, whereas cells with impaired apoptotic pathway caused by defective FADD/BID axis develop resistance to T-cell mediated immunotherapy and result in tumor progression (created with Biorender.com).

LLC). Cell death was calculated by including both early and late apoptotic cells.

**Cytokine quantification assay**. For determination of functional profiles of CARTs, MM cells were co-cultured with BCMA-CART cells (4:1 effector to target ratio) for 20 h. Cells were also incubated with 0.1 µg/ml of daratumumab or 1 nM teclistamab along with (4:1 E:T ratio) PBMCs or Pan-T cells from healthy donors for 20 h. Supernatant was collected and cytokine quantification was performed using ELISA MAX deluxe set Human IFN-γ (430116) and Human IL-2 (431816) (Biolegand) according to manufacturers' instructions.

**Drug testing**. MM cells were seeded in 96-well plates (20,000 cells/well) and incubated with respective compounds (melphalan, bortezomib, doxorubicin, belantamab mafodotin) for 3 days and with lenalidomide for 5 days and subjected to alamarBlue assay. Cells were treated with 0.1 µg/ml daratumumab or 1 nM teclistamab along with (5:1 or 10:1 effector: target ratio) PBMCs isolated from healthy donors. Cell viability was determined using bioluminescence signal.

**Caspase activity assay**. Cells were treated with 15–20 µg/ml of Fas Ab (Merck) for 24 h and caspase activities were measured using Caspase-Glo 3/7 Assay and Caspase-Glo 8 Assay (Promega) according to manufacturer's instruction.

**dSTORM imaging**. For reversible photoswitching of Alexa Fluor 647 (AF647), a PBS-based imaging buffer (pH 7.4) was used that contained 100 mM Cysteamine hydrochloride (M6500-25G, Sigma). dSTORM measurements were performed[19,20] by using an Olympus IX-71 inverted microscope (Olympus) equipped with an oil immersion objective (PlanApo N 60 × 1.45 TIRF, Olympus) and a nosepiece stage (IX2-NPS, Olympus). AF647 was excited with a laser (iBeam smart 640-S Toptica, Photonics AG). The excitation light was spectrally cleaned by an appropriate bandpass filter (642/10, Semrock) and then focused onto the back focal plane of the objective. To switch between different illumination modes (EPI, HILO, and TIRF illumination), the lens system and mirrors were arranged on a linear translation stage. The fluorescence emission was collected by the same objective and transmitted by the dichroic beam splitter (FF545/650-Di01, Semrock). The emission light was filtered by a long pass filter (647 nm RazorEdge, Semrock) before being projected onto an electron-multiplying CCD camera (iXon DU-897D-CS0-BV, Andor Technology Ltd). A final pixel size of 133 nm was generated by placing additional lenses in the detection path. 15,000 frames were recorded with a frame rate of ~50 Hz (20 ms exposure time) and an excitation intensity of ~3 kW/cm2.

From the recorded image, a table with all localizations as well as a reconstructed dSTORM image were generated using the single-molecule localization software rapidSTORM 3.3[21]. For analysis of each dSTORM image an appropriate region of interest at the basal membrane of the cell, was chosen using the Napari viewer implemented in the analysis tool box LOCAN [Available from: https://zenodo.org/record/5722473]. Afterward, clustering of localization data was performed using a DBSCAN[22,23] algorithm with the parameters ε = 20 nm and minimum points (MinPts) = 3. The final clustering data was depicted as box plots using the software OriginPro 2021b [Available from: https://www.originlab.com/index.aspx?go=Company&pid=1130].

**Western blotting**. Frozen cell pellets were lysed in 1x RIPA lysis buffer supplemented with Halt protease inhibitor cocktail (ThermoFisher Scientific). Samples were mixed with Laemmlie-

buffer, heated at 90 °C for 3 min before running on NuPAGE Tris-Acetate 3–8% gel (ThermoFisher Scientific). Wet blotting was performed using Xcell SureLock mini-Cell electrophoresis system. Primary antibodies used were cleaved-PARP (Cell signaling) and GAPDH (Cell signaling) and secondary antibody was anti-rabbit IgG, HRP-linked (Cell signaling). Blots were developed using ESL (BioRad). Images were taken using Chemidoc (Biorad).

**Bulk RNA seq**. Total RNA was isolated from bone marrow cells using manufacturers instruction (Qiagen). For transcriptome analysis, 250 ng of total RNA per sample were used to produce stranded RNA libraries (TruSeq Total Stranded RNA, Illumina). NovaSeq 6000 (Illumina) was used to 2x100bp paired-end with a median of 64 million reads per sample. Reads were aligned (STAR, version 2.5.0a) to the human reference genome (hg19) and Cufflinks (version 2.2.1) was used to estimate gene counts. Data normalization was performed and $Log_2TPM$ was calculated.

**qPCR for gene expression analysis**. Total RNA was isolated using RNeasy Mini Kit (Qiagen) following manufacturers instruction. cDNA was synthesized using SuperScript VILO cDNA-Synthesiskit (ThermoFisher Scientific). Gene expression analysis was performed using TaqMan Real-Time-PCR-Assays (ThermoFisher Scientific) using *GUS4B* or *GAPDH* as housekeeping genes.

**Patient analysis**. Patient demographics, MM-related data, cytogenetics, and pretreatments were investigated. All ethical regulations relevant to human research participants were followed. Informed consent was obtained from all patients in our study. The study was approved by the Internal Review Board at the University hospital of Würzburg (Wü8/21) and adhered to the tenets of the Declaration of Helsinki. The characteristics of the 22 patients being resistant to daratumumab are presented in Supplementary Table 1.

**Statistics and reproducibility**. All in vitro data presented here for drug testing are representative of at least three independent replicates. Comparison between two groups was performed using unpaired *t*-test and comparison of more than two groups was performed using two-way ANOVA. Fischer's exact test was performed for daratumumab resistance analysis in patient cohort with high and low expression of target genes. All results are represent mean values ± SEM.

**Reporting summary**. Further information on research design is available in the Nature Portfolio Reporting Summary linked to this article.

## Data availability
Guide RNA sequences and $Log_2TPM$ values are provided as Supplementary Tables (1 and 2). Uncropped western blot are available as Figure S4. The source data behind the graphs in the paper are available in Supplementary Data. RNA sequencing data is available at following accession number (https://doi.org/10.5281/zenodo.10261940). Other data generated in this study are available from corresponding authors upon request.

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

## Acknowledgements

K.M.K. was supported by "Stiftung zur Förderung der Krebsforschung an der Universität Würzburg", the "Stifterverband", the "CDW Stiftung" and "Janssen". K.M.K. and L.R. were funded by German Cancer Aid (Deutsche Krebshilfe) via the "Mildred Scheel Early Career Center" (MSNZ) program.

## Author contributions

Concept and design was developed by U.M., L.R., and K.M.K. Experiments and data acquisition was performed by U.M., X.Z., S.P., S.N., C.V., P.E., N.B., M.J.S., M.T., J.M., E.C., E.T., S.H., L.H., M.A.N.A., A.O.R., A.F.M. Data analysis and interpretation of Data was performed by U.M. L.R., K.M.K., M.G., A.V., M.H., C.H., S.D., and M.S.; U.M., X.Z., J.W., L.R., K.M.K. wrote the manuscript and P.R., M.R., J.M.L., H.E. corrected and approved the manuscript.

## Funding

## Competing interests

The authors declare the following competing interests: K.M.K. reports other support from Stifterverband für die Deutsche Wissenschaft, personal fees from Celgene, BMS, AbbVie, GSK, and Takeda; grants and personal fees from Janssen; and grants from SkylineDx and German Cancer Aid—MSNZ. C.H. reports other support from MLL Munich Leukemia Laboratory. J.M.-L. reports personal fees and nonfinancial support from Janssen, Sanofi, BMS, Roche and Gilead. H.E. reports grants and other support from Janssen, BMS/Celgene, Amgen, GSK, and Sanofi, as well as other support from Takeda and Novartis. L.R. reports personal fees from BMS, Janssen, Pfizer, Amgen, GSK, Sanofi and support from German Cancer Aid—MSNZ. Other authors declare no competing interests.
