## [Peer Review File · Communications Biology]

Reviewers' comments:

Reviewer #1 (Remarks to the Author):

In this manuscript, Munawar and colleagues report that resistance to cell death receptor-mediated (FAS/BID axis) killing drives cross-resistance to immunotherapy in multiple myeloma. Conceptually, it has been well documented that T-cell mediated killing is FAS/FADD/TNF-R dependent, study from these authors still provide additional evidence to this concept by using multiple myeloma as a model system.

Based on the findings from the study, it is convincing that FAS/BID knockdown protects two MM cell lines from CART-mediated killing. However, the data supporting the impairment of death receptor signaling due to resistance to immunotherapy is rather weak (Figure 4A, no significance). Hence, it is acceptable to state that FAS-BID is required to induce cell killing resulting from immunotherapy. But impairment in cell death receptor pathway is a mediator of resistance to immunotherapy is an overstatement and not supported by present data.

Figures 2 and 3 are missing untransduced T cell and non-BCMA cell line as negative controls. It is required to show that the effects is specific to CAR and target antigen.

Reviewer #2 (Remarks to the Author):

Summary of the study

In this manuscript, Munawar et al. proposed that resistance displayed by MM cells against immunotherapy is mediated by impaired death receptor signaling. It had been previously demonstrated that aberrant death receptor signaling is the cause of immunotherapy resistance in ALL. In this manuscript, the authors investigated whether impaired receptor signaling can also be the cause of resistance in MM. To address this, the authors deployed known MM cell lines AMO-1 and L363 and generated FADD and BID KO based on this cell line to study their response to immunotherapies and classical chemotherapeutic agents. They show that FADD/BID-deficient cells had reduced response to immunotherapy-based drugs compared to WT. However, classical genotoxic drugs induced similar responses in the two groups.

Accompanying the presented in vitro data, the authors report FADD and BID expression levels in MM patients. They report that patients with the lowest FADD and BID were more daratumumab-resistant. In contrast, patients with the highest FADD and BID expression were less daratumumab resistant, possibly supporting the physiological role of the extrinsic death machinery in immunotherapy-mediated cell death.

Major points

1) Reconstitution study by expressing (if required, CRISPR-resistant) FADD/BID in the KO background to rescue the phenotype and to confirm the specificity of the deletion would strengthen in vitro observations.

2)The authors proposed that the inefficacy of immunotherapies against MM cells is mediated by impaired death receptor signaling caused potentially by low expression of FADD/BID. However, the authors found similar FADD and BID expression levels between NDMM and RRMM. While the authors

mention this could be due to other resistance mechanisms, they did not provide evidence. Analysis of key known regulators of the extrinsic death pathways, such as cFLIP, for example, should be included in the analysis.

3)The resolution of the microscopy images for surface antigens should be improved.

Minor points

Line (112), unnecessary repeat of "in WT cells" in the sentence.

Reviewer #1 (Comment 1):

Based on the findings from the study, it is convincing that FAS/BID knockdown protects two MM cell lines from CART-mediated killing. However, the data supporting the impairment of death receptor signaling due to resistance to immunotherapy is rather weak (Figure 4A, no significance). Hence, it is acceptable to state that FAS-BID is required to induce cell killing resulting from immunotherapy. But impairment in cell death receptor pathway is a mediator of resistance to immunotherapy is an overstatement and not supported by present data.

Response: *We thank reviewer 1 for the comment. We have toned down our statement on the global role of cell death receptor signaling pathway and focused rather on the FADD/BID axis to be responsible for diminished cell killing in the context immunotherapy resistance. Based on this comment we would like to kindly suggest to change the title of our manuscript to “Impaired FADD/BID signaling mediates cross-resistance to immunotherapy in Multiple Myeloma“. For the remaining parts in the manuscript, we have changed the following sections accordingly:*

Line 122 (heading):

*R0 version: **Death receptor signaling** regulates resistance to T-cell mediated immunotherapy and is dispensable for conventional anti-MM therapies*

*R1 version: **FADD / BID axis** regulates resistance to T-cell mediated immunotherapy and is dispensable for conventional anti-MM therapies*

Line 124:

*R0 version: To validate the significance of **Death receptor signaling***

*R1 version: To validate the significance of **FADD/BID***

Line 149:

*R0 version: In summary, these findings confirm that **defective extrinsic apoptotic** circuitry conferred by *FADD* and *BID* knock-out results in reduced activity of T-cell mediated immunotherapies in MM, whereas conventional anti-MM agents remain equally effective.*

*R1 version: In summary, these findings confirm that **defects in extrinsic apoptotic circuitry conferred by FADD and BID** knock-out results in reduced activity of T-cell mediated immunotherapies in MM, whereas conventional anti-MM agents remain equally effective.*

Reviewer #1 (Comment 2):

Figures 2 and 3 are missing untransduced T cell and non-BCMA cell line as negative controls. It is required to show that the effects is specific to CAR and target antigen.

Response: *We thank reviewer 1 for this important comment and have included the negative control of untransduced T cells (labelled as Mock) in Figure 2 and 3.*

Figure 2

Figure 2) FADD/BID axis regulates T-cell mediated immunotherapies. Luminescence-based cell survival assay for WT, FADD^{KO} and BID^{KO} AMO1 (A) and L363 (B) cells lines treated with daratumumab and co-cultured with PBMCs from healthy donors. Target cells (AMO1 (C)) (L363 (D)) were mixed with BCMA-CARTs or untransduced T cells (Mock) in 10:1 E:T ratio for FADD^{KO} and 5:1 E:T ratio for BID^{KO} cells and were subjected to the luminescence-based cell survival assay. L363 WT and FADD^{KO} cells were co-cultured with Pan-T cells isolated from healthy donors in the presence of teclistamab and viability was determined by luminescence-based cell survival assay (E). *= p<0.05; **= p<0.005; ***= p<0.0005

Figure 3

Figure 3) Loss of target antigen or dysfunction of (CAR-)T cells are not responsible for therapeutic resistance. (A) Cells were stained for CD38 (Top panel) BCMA (middle panel) and CD138 (bottom panel) and imaged by dSTORM. No differences in clusters/ μm^2 for mentioned antigens was observed. BCMA-CARTs or (B) and Pan-T cells (C) from two different donors were stimulated with WT (AMO1-L363) or respective KO cells for 20hr and levels of IL-2 and IFN- γ were measured. Mock represents the untransduced T cells (B) and Control represents the base level of IL-2 and IFN- γ in absence of Teclistamab (C).

Reviewer 1 raises an important aspect and we agree that a non-BCMA cell line could be a control for this experiment. BCMA is expressed on all multiple myeloma cell lines. Using a non-myeloma cell line without BCMA expression may not be an appropriate control for the trajectory of this study. As in this study various antigens are targeted with different immunotherapeutic agents, we decided to use a high-resolution dSTORM microscopy approach to determine surface antigen expression for all of these targets. We kindly ask for understanding that the generation of knock-out MM cell line models for each of these target antigens was outside the scope of this study.

Reviewer #2 (Comment 1):

Reconstitution study by expressing (if required, CRISPR-resistant) FADD/BID in the KO background to rescue the phenotype and to confirm the specificity of the deletion would strengthen in vitro observations.

Response: We thank reviewer 2 for this important comment and agree on the need of a rescue experiment to further strengthen our in vitro observations. For the R1 version of our manuscript, we have now performed a rescue experiment and found that KO cells are not able to cope with the plasmid (pc.DNA3.1)-mediated expression of FADD and BID. qPCR analysis after re-expression of FADD and BID revealed a high level compared to WT base line (Figure 1 A, B). Cells were not able to deal with the high level of the FADD and BID expression and therefore showed high levels of cell death (Figure 1 C, D) which makes it impossible to perform downstream experiments. Furthermore, our observation fall in line with some prior studies which claim that re-expression of FADD and BID promotes the apoptosis in the cells. Wang and colleagues (Wang HB et al. *Biomed Pharmacotherapy*, 2017, PMID 28618251) have shown that the overexpression of FADD in glioblastoma cells promotes apoptosis. Other groups have shown activation of apoptosis in response to BID overexpression in non-small cell lung cancers (Fukazawa T et al. *Journal of Biological Chemistry*, 2003, PMID: 12690107) and ovarian cancer stem cells (Long Q et al. *Oncology Reports*, 2017, PMID 27878291).

Figure 1

Reviewer #2 (Comment 2):

The authors proposed that the inefficacy of immunotherapies against MM cells is mediated by impaired death receptor signaling caused potentially by low expression of FADD/BID. However, the authors found similar FADD and BID expression levels between NDMM and RRMM. While the authors mention this could be due to other resistance mechanisms, they did not provide evidence. Analysis of key known regulators of the extrinsic death pathways, such as cFLIP, for example, should be included in the analysis.

Response: We thank reviewer 2 for the comment. We have extended the analysis to other key players of the extrinsic apoptotic pathway.

Line 172:

R0 version: To determine the clinical relevance of our findings we screened for **decreased FADD and BID** expression levels in a pilot cohort of n=24 newly diagnosed MM (NDMM) and n=54 relapsed refractory MM (RRMM) patients without prior exposure to TCE and/or CART cell therapy.

R1 version: To determine the clinical relevance of our findings we screened for expression levels **key players of apoptotic machinery** in a pilot cohort of n=24 newly diagnosed MM (NDMM) and n=54 relapsed refractory MM (RRMM) patients without prior exposure to TCE and/or CART cell therapy.

Line 176:

R0 version: We did not find significant expression differences at cohorts level.

R1 version: We did not find significant differences in expression levels for Caspase 3,7,8,9 and c-FLIP across cohorts in NDMM vs. RRMM patients (Figure S3)

Data is provided as supplemental figure S3. Furthermore, we have adapted our statement in response to reviewer 1 to specifically elaborate on the role of FADD/BID rather than the whole cell death receptor pathway.

Figure S3

Figure S3) Gene expression data of Caspase 3 (A) c-FLIP (B) Caspase 9 (C) Caspase 7 (D) and Caspase 8 (E) in healthy donors (HD), NDMM and RRMM patients. Gene expression was determined by bulk RNA seq and normalized as $\log_2 \text{TPM}$ and quantified by qPCR and normalized as $2^{\Delta CT}$.

Reviewer #2 (Comment 3):

The resolution of the microscopy images for surface antigens should be improved.

Response: We thank reviewer 2 for pointing out the resolution of images. Image quality has been improved in the R1 version of this manuscript. If needed, individual high definition images may also be provided as supplemental data in the appendix.

Figure 3

Reviewer #2 (Comment 4):

Line (112), unnecessary repeat of “in WT cells” in the sentence.

Response: *We thank reviewer 2 this remark and corrected the phrasing accordingly.*

Reviewers' comments:

Reviewer #1 (Remarks to the Author):

The following claim is not supported by existing data shown in Figure 4A. There is not statistical difference between ND vs R/R patients in the expression of BID/FADD or other downstream effector caspases.

"With correlative clinical data indicating that the FADD/BID axis is indispensable for meaningful responses to novel immunotherapies in vivo, we report impaired death receptor signaling as a novel resistance mechanism to T-cell mediated immunotherapy in MM."

Reviewer #2 (Remarks to the Author):

The authors have addressed the majority of the comments from reviewers, which has resulted in enhanced data quality and interpretation. The manuscript elucidates how the cell death pathway may contribute to resistance mechanisms against T-cell mediated immunotherapy.

Reviewer #1:

The following claim is not supported by existing data shown in Figure 4A. There is not statistical difference between ND vs R/R patients in the expression of BID/FADD or other downstream effector caspases.

"With correlative clinical data indicating that the FADD/BID axis is indispensable for meaningful responses to novel immunotherapies in vivo, we report impaired death receptor signaling as a novel resistance mechanism to T-cell mediated immunotherapy in MM."

Response: *We thank reviewer 1 for the comment. We have modified our manuscript and discussed the limitations of clinical implications of our cell models.*

Line 60:

R1 version: This resistance seems exclusive to novel immunotherapies while sensitivity to conventional anti-tumor therapies being preserved. **With correlative clinical data indicating that the FADD/BID axis is indispensable for meaningful responses to novel immunotherapies in vivo,** we report impaired death receptor signaling as a novel resistance mechanism to T-cell mediated immunotherapy in MM.

R2 version: This resistance seems exclusive to novel immunotherapies while sensitivity to conventional anti-tumor therapies being preserved **in vitro. As a proof of concept, we present a confirmatory clinical case indicating that the FADD/BID axis is required for meaningful responses to novel immunotherapies thus,** we report impaired death receptor signaling as a novel resistance mechanism to T-cell mediated immunotherapy in MM.

Line 215:

R1 version: **Interestingly we did not observe** differing mean levels of key apoptotic genes between NDMM vs. RRMM at cohort level. However, pronounced tendency of Daratumumab resistance in patients with low expression of FADD/BID accentuate their importance in immunotherapy resistance.

R2 version: **However, in our retrospective real world analysis** we did not observe differing mean levels of key apoptotic genes between NDMM vs. RRMM, **most likely due to the variety of different combination therapies that the patients received. Nevertheless, in our Daratumumab resistant RRMM patients a clear trend towards low expression of FADD/BID was visible.**

Reviewer #2:

The authors have addressed the majority of the comments from reviewers, which has resulted in enhanced data quality and interpretation. The manuscript elucidates how the cell death pathway may contribute to resistance mechanisms against T-cell mediated immunotherapy.

Response: We thank reviewer 2 for the insightful remarks which have been substantial to improve the quality of our work.